# Incidence and Risk Factors for Development of Cardiac Toxicity in Adult Patients with Newly Diagnosed Acute Myeloid Leukemia

**DOI:** 10.3390/cancers15082267

**Published:** 2023-04-12

**Authors:** Blanca Boluda, Antonio Solana-Altabella, Isabel Cano, David Martínez-Cuadrón, Evelyn Acuña-Cruz, Laura Torres-Miñana, Rebeca Rodríguez-Veiga, Irene Navarro-Vicente, David Martínez-Campuzano, Raquel García-Ruiz, Pilar Lloret, Pedro Asensi, Ana Osa-Sáez, Jaume Aguero, María Rodríguez-Serrano, Francisco Buendía-Fuentes, Juan Eduardo Megías-Vericat, Beatriz Martín-Herreros, Eva Barragán, Claudia Sargas, Maribel Salas, Margaret Wooddell, Charles Dharmani, Miguel A. Sanz, Javier De la Rubia, Pau Montesinos

**Affiliations:** 1Hematology Department, Hospital Universitari i Politècnic La Fe, 46026 Valencia, Spainirene_navarro@iislafe.es (I.N.-V.);; 2Instituto de Investigación Sanitaria La Fe, 46026 Valencia, Spain; 3Pharmacy Department, Hospital Universitari i Politècnic La Fe, 46026 Valencia, Spain; 4CIBERONC, Instituto de Salud Carlos III, 28029 Madrid, Spain; 5Cardiology Department, Hospital Universitari i Politècnic La Fe, 46026 Valencia, Spain; 6Daiichi Sankyo, Inc., Basking Ridge, NJ 07920, USA; 7Center for Real-World Effectiveness and Safety of Therapeutics (CREST), University of Pennsylvania Perelman School of Medicine, Philadelphia, PA 19104, USA; 8Internal Medicine, School of Medicine and Dentistry, Catholic University of Valencia, 46001 Valencia, Spain

**Keywords:** acute myeloid leukemia, cardiac toxicity, risk factors, real-life

## Abstract

**Simple Summary:**

The incidence of cardiac morbimortality in acute myeloid leukemia (AML) is not well known. We aim to estimate the cumulative incidence (CI) of cardiac events in AML patients and to identify risk factors for their occurrence. We observed a high incidence of cardiac events (58.5%) among 525 treated patients, coupled with significant mortality due to cardiotoxicity (3.6%). The presence of relevant cardiac antecedents was the main risk factor for developing fatal cardiac events (hazard ratio (HR) = 1.9). Age ≥ 65 (HR = 2.2), relevant cardiac antecedents (HR = 1.4), and non-intensive chemotherapy (HR = 1.8) were associated with non-fatal cardiac events. We observed that, among 285 intensive therapy patients, median overall survival was decreased in those experiencing grade 3–4 cardiac events (*p* < 0.001). We identified prognostic factors that increase the risk of cardiac events, which may be useful in selecting high-risk patients for stringent cardiac monitoring and management.

**Abstract:**

The incidence of cardiac morbimortality in acute myeloid leukemia (AML) is not well known. We aim to estimate the cumulative incidence (CI) of cardiac events in AML patients and to identify risk factors for their occurrence. Among 571 newly diagnosed AML patients, 26 (4.6%) developed fatal cardiac events, and among 525 treated patients, 19 (3.6%) experienced fatal cardiac events (CI: 2% at 6 months; 6.7% at 9 years). Prior heart disease was associated with the development of fatal cardiac events (hazard ratio (HR) = 6.9). The CI of non-fatal cardiac events was 43.7% at 6 months and 56.9% at 9 years. Age ≥ 65 (HR = 2.2), relevant cardiac antecedents (HR = 1.4), and non-intensive chemotherapy (HR = 1.8) were associated with non-fatal cardiac events. The 9-year CI of grade 1–2 QTcF prolongation was 11.2%, grade 3 was 2.7%, and no patient had grade 4–5 events. The 9-year CI of grade 1–2 cardiac failure was 1.3%, grade 3–4 was 15%, and grade 5 was 2.1%; of grade 1–2, arrhythmia was 1.9%, grade 3–4 was 9.1%, and grade 5 was 1%. Among 285 intensive therapy patients, median overall survival decreased in those experiencing grade 3–4 cardiac events (*p* < 0.001). We observed a high incidence of cardiac toxicity associated with significant mortality in AML.

## 1. Introduction

Despite the progress in modern chemotherapy achieved in recent decades, the prognosis of acute myeloid leukemia (AML) remains discouraging, with a considerable proportion of patients suffering induction failure or relapse after attaining complete remission (CR) [1,2]. AML patients receive potentially cardiotoxic drugs such as anthracyclines [3,4,5,6,7] and drugs with corrected QT interval (QTc) prolongations, such as FLT3 inhibitors [8,9,10,11] and azoles, among others [12,13]. Cardiotoxicity is known to occur among all cancer survivors; the United States (US) National Health and Nutrition Examination Survey, performed on 1807 cancer survivors and followed up for seven years, showed that up to 33% of the patients died of heart disease [14]. Nevertheless, there is still no ’standard‘ definition of cardiotoxicity [4,15], and the incidence of cardiac morbidity and mortality in the AML population is not well known. In addition to the potential toxicity of chemotherapy, cardiac events can emerge in patients with previous cardiac diseases, due to other concomitant medications, or as a complication of AML itself [3,5,16]. So far, no previous studies have provided a holistic view of cardiac toxicities in a real-world series of AML patients.

The aim of this retrospective chart review study is to provide insights into the overall incidence and risk factors for the occurrence of cardiac events in a large series of unselected adult AML patients in a single Spanish institution (Hospital Universitari I Politècnic La Fe).

## 2. Methods and Patients

### 2.1. Study Design and Population

This was a single-center, non-interventional, retrospective, systematic chart review study. The study population included adult patients diagnosed with AML during the study period (from 1 January 2011 to 30 June 2020) at the Hospital Universitari i Politècnic La Fe (HULaFe). Patients diagnosed with acute promyelocytic leukemia (APL) were excluded. AML patients who were at least 18 years old at diagnosis (i.e., index date) and had charts available for exhaustive review and data collection were included. Patients who participated in any interventional clinical trial from the index date were not excluded (i.e., with FLT3 inhibitors).

The protocol was approved by the local Clinical Research Ethics Committee of HULaFe, and patients provided informed consent in accordance with the principles of the Declaration of Helsinki.

The study population was made up of all adult patients with AML who were diagnosed at the HULaFe (overall cohort). The first-line (1L) cohort included treated patients who were part of the overall cohort, but they differed from the overall cohort in their follow-up period; overall cohort patients were followed until the last follow-up recorded or death, while 1L patients were followed until relapse/refractory (R/R) or death date.

### 2.2. Study Objectives and Variables

The primary objective was to describe the overall incidence of fatal cardiac events in AML patients from the primary diagnosis date at the HULaFe until the last follow-up or death. The secondary objective was to describe the overall incidence of non-fatal cardiac events (including life-threatening ones) in AML patients from the primary diagnosis date until the last follow-up or death.

Other objectives included (1) establishing risk factors for the development of cardiac events in AML; (2) determining the overall incidence of cardiac events in all AML patients from the primary diagnosis date until the end of the follow-up or the first R/R date during the front-line period (1L cohort); (3) establishing risk factors for the development of cardiac events during the front-line period (1L cohort); and (4) describing response rates and long-term outcomes (overall survival (OS), event-free survival (EFS)) in patients with or without cardiac events from the index date until the last follow-up.

The following information was collected at baseline when available: (1) Patient and disease characteristics, including age, gender, cardiac and other co-morbidities; French–American–British (FAB) classification, World Health Organization (WHO) classification [17], Eastern Cooperative Oncology Group (ECOG) scale, de novo or secondary AML, bone marrow assessment (blasts percentage), extramedullary involvement, cytogenetic risk stratification, gene mutations (FLT3-ITD, FLT3-TKD, NPM1), peripheral blood parameters, date of initial diagnosis, baseline cardiologic medication, prior treatment with anthracyclines. (2) Therapy-related variables: treatment regimen for AML; hematopoietic stem cell transplantation (HSCT) and date if performed; type of HSCT; investigational therapy in the first-line; FLT3 inhibitors; response assessment (complete remission (CR), complete remission with incomplete hematologic recovery (Cri), partial remission (PR), morphologic leukemia-free state (MLFS), disease progression/relapse, non-evaluable); date of response assessment; date of death; relapse; date of relapse; date of last follow-up. The following variables related to cardiac events were collected: date of cardiac event, cardiac event term (according to Common Terminology Criteria for Adverse Events Version 5.0 (CTCAE V5)) [18], grade, admission for an event (inpatient/outpatient, domiciliary hospitalization), level of care during hospitalization (ICU, hematology or cardiology), outcome of cardiac events (resolved/unresolved), concomitant medications, and causes contributing to cardiac events.

The systematic review of the patients’ medical records included the revision of all electrocardiograms (ECGs) available in the electronic clinical records by trained hematologists, with cardiologist consultation if needed; a systematic review of all clinical notes from diagnosis; and vital signs, focusing on heart rate and blood pressure. Deaths after a cardiac event were attributed to the cardiac event or to another cause (i.e., sepsis or AML progression) according to the investigator’s clinical judgment.

### 2.3. Definitions

The diagnosis date refers to the date of AML diagnosis. Refractory AML was defined as a failure to achieve CR/CRi after 1 or 2 courses of intensive-induction chemotherapy treatment [19,20], excluding patients with death in aplasia or death due to indeterminate cause. Patients treated with non-intensive chemotherapy approaches were considered refractory when they showed progressive disease, no clinical benefit, or a change in the treatment line. Relapse was defined as AML subjects who achieved a CR/CRi with prior line treatment and had a hematologic relapse. Treatments were classified as intensive chemotherapy (i.e., idarubicine and cytarabine (3 + 7); fludarabine–cytarabine-based (FLAG-based) or high- or intermediate-dose cytarabine-containing regimens) or non-intensive approaches (i.e., hypomethylating agents, low-dose cytarabine-based regimens (LDAC), FLT3 inhibitors as monotherapy).

Regarding cardiac antecedents, any prior history of cardiac disease was recorded, including arterial hypertension. We defined a category of clinically relevant cardiac antecedents including conditions with worse prognosis and severity: symptomatic heart failure, myocardial infarction, or ejection fraction (EF) < 50%. Several categories of cardiac events were analyzed: (1) myocardial ischemic events included the following (according to CTCAE): cardiac troponin increased, chest pain (cardiac), and myocardial infarction; (2) heart failure and related events included ejection fraction decreased, heart failure, left ventricular systolic dysfunction, pulmonary edema, and right ventricular dysfunction; (3) arrhythmia events included atrial fibrillation, atrial flutter, complete atrioventricular (AV) block, first degree AV block, Mobitz type II AV block, Mobitz type I AV block, paroxysmal atrial tachycardia, sick sinus syndrome, supraventricular tachycardia, torsade de pointes, ventricular arrhythmia, ventricular fibrillation, ventricular tachycardia, sudden death, cardiac arrest, and cardiorespiratory arrest. An isolated QTc prolongation was not considered in the category of arrhythmia events. The QTc interval was calculated using the Fridericia formula (QTcF). Echocardiograms for the ejection fraction were performed at diagnosis in patients treated with intensive chemotherapy and for unfit patients when clinically indicated or required by clinical trial; during follow-up, an echocardiogram was performed previous to stem cell transplant (both autologous and allogeneic) or when clinically indicated.

Life-threatening events were defined as those requiring an intensive care unit (ICU) admission or intensive management in a hematology unit as per clinical judgment.

### 2.4. Statistical Analyses

Descriptive statistics were calculated for all outcomes. Means or medians were provided for continuous variables and incidences and proportions for categorical variables. The crude incidence (rate or percentage) of patients developing a fatal cardiac event (number of fatal cardiac events/total number of patients for each study cohort) was calculated. Categorical variables were summarized by the number and proportion in each category. Chi-square with Yates’ correction, Mann–Whitney U, and Student’s t-tests were used to analyze differences in the distribution of variables among patient subsets. The proportion of subjects with missing data for key variables collected in the study was described in the tables as a separate category. Data collection methods (single-center) implemented in this study ensured that missing data were minimized.

For non-fatal and overall cardiac events, as patients could develop such complications several times during their observation period, the crude incidence (rate or percentage) was calculated accounting for the first cardiac event for each patient. For the univariate risk factor analyses (using the Fine and Gray method), we considered a significant correlation for *p*-values < 0.05. The cumulative incidence was also calculated (using the cumulative incidence method, as described by Fine and Gray) [21] and the first cardiac event was counted as an event (according to the defined category, i.e., fatal, non-fatal). Patients presenting with several cardiac events were considered to have only one event for the purposes of calculating overall incidence. In patients with several cardiac events, we considered the most severe cardiac event as the first (i.e., in a case of 2 events, with the first being a grade 1–2 and then a different grade 3–4 event, the latter was considered the event for the purposes of calculations). Time-to-event analyses were calculated in months (from the first treatment after the index date) and were summarized with Kaplan–Meier (KM) curves. For OS, an event was defined as death by any cause, while for EFS, it was failure to achieve CR/CRi (counted on day 1 after starting R/R therapy), relapse after CR/CRi, or death by any cause, whichever occurred first. The impact of risk factors for cardiac events was assessed using multivariate Cox proportional hazard regression. Analyses were conducted in June 2022 using R.2.14 statistical software for all calculations.

## 3. Results

### 3.1. Patient Disposition and AML Characteristics

Overall, 571 adult consecutive patients with AML were diagnosed and/or treated at the HULaFe from January 2011 to June 2020. Overall, 46 patients received only supportive care, and 525 patients received various front-line regimens. In total, 285 patients received intensive chemotherapy: 218 (38%) received standard intensive chemotherapy, and 67 (12%) received intensive clinical trials. In total, 240 patients received non-intensive chemotherapy: 7 (1%) received hypomethylating agents, 68 (13%) received LDAC-based regimens, and 165 (31%) underwent non-intensive clinical trials (Table 1).

The median age of patients in the overall cohort (n = 571) was 65 years (range, 18–98 years) (Appendix A); 331 (58%) were male; ECOG was >2 in 58 (10%); 350 (61%) were de novo AML and 221 (39%) were secondary or therapy-related AML; 77 (14.7%) had the FLT3-ITD mutation; and 38 (7.2%) received front-line FLT3 inhibitors. Up to 14% of patients had antecedent or ongoing relevant cardiac comorbidities at the time of AML diagnosis, and 38% had cardiac comorbidity (Appendix A). Patients with relevant cardiac comorbidities were more frequently older than 65 years (70% vs. 49%, *p* = 0.001) and male (73% vs. 55%, *p* = 0.004) and more frequently had an ECOG ≥ 2 (34% vs. 23%, *p* = 0.04) and increased serum creatinine levels (29% vs. 13%, *p* < 0.001) (Table 2). Intensive chemotherapy was less frequently administered to patients with cardiac comorbidities (23%) as compared with those without cardiac comorbidities (41%) (*p* = 0.043). Differences between patients receiving intensive chemotherapy versus other approaches are shown in Appendix A.

### 3.2. Fatal Cardiac Events in the Overall Cohort

Overall, 26 fatal cardiac events were recorded among 571 patients (crude incidence of 4.6%). The crude incidence of patients experiencing a fatal cardiac event among 46 untreated patients was 15.2% (7 out of 46).

Among 525 treated patients, 19 experienced fatal cardiac events (crude incidence of 3.6%); there was a CI of 2% at 6 months and a CI of 6.7% at 9 years (Figure 1A). Among fatal cardiac events, one was due to an aortic valve rupture (previous endocarditis), one was a cardiac tamponade, eleven were due to heart failure (three of them occurred before starting treatment and three occurred in the follow-up phase), and six were categorized as arrhythmia, although no acute ECG abnormality was found (four cardiorespiratory arrests, one sudden death, and one asystole). In the univariate analysis, patients with prior cardiac antecedents had an increased incidence of fatal cardiac events compared with patients with no prior cardiac disease (CI at 9 years, 20.1% vs. 4.9%; *p* < 0.001) (Table 3). Multivariate analysis showed that the presence of relevant cardiac antecedents was associated with an increased incidence of fatal cardiac events (hazard ratio (HR), 6.9; confidence interval, 95% (CI 95%), 2.8–16.7; *p* < 0.001] (Appendix A).

### 3.3. Non-Fatal Cardiac Events in the Overall Cohort

The CI of non-fatal cardiac events was 43.7% at 6 months and 56.9% at 9 years (Figure 1B). The 9-year CI of grade 3–4 events was 42.6% and 20.6% for grades 1–2. In the univariate analysis, age ≥ 65 years old (64.4% vs. 49.9%, *p* < 0.001), relevant cardiac antecedents (73% vs. 54.6%, *p* = 0.004), and inclusion in a front-line clinical trial (65.2% vs. 50.8%, *p* < 0.001) were associated with an increased CI of non-fatal cardiac events. Age ≥ 65 (HR 2.2, CI 95% 1.5–3.3, *p* < 0.001), relevant cardiac antecedents (HR 1.4, CI 95% 1.4–2, *p* = 0.02), and non-intensive chemotherapy as a front-line treatment (HR 1.8, CI 95% 1.2–2.8, *p* = 0.004) were associated with an increased CI of non-fatal cardiac events in the multivariate analysis (Appendix A).

The CI of life-threatening cardiac events was 4.1% at 6 months and 6.4% at 9 years. They occurred more frequently among patients not included in clinical trials and those treated with intensive approaches (not statistically significant) (Table 4).

### 3.4. Timing of Cardiac Events

Overall, 488 cardiac events occurred in 307 patients during the observation period. Among the 307 patients with at least 1 cardiac event, 38 (12%) developed the first event before starting treatment, 132 (43%) during the first cycle, 31 (10%) during consolidation, 55 (18%) during non-intensive further cycles, 14 (5%) during HSCT, and 37 (12%) during follow-up (Figure 2, Appendix A; Appendix A).

### 3.5. Cardiac Events in Intensive Versus Non-Intensive Front-Line Cohorts

Among 285 patients treated with intensive chemotherapy, the CI of fatal events was 1.9% at 6 months and 6.3% at 9 years, and the CI of non-fatal events was 42.5% at 6 months and 54.7% at 9 years (Appendix A). In the univariate analysis, patients with relevant cardiac antecedents and previous treatment with anthracyclines had an increased 9-year CI of fatal cardiac events. Patients aged ≥ 65 years, with relevant cardiac antecedents, and included in the front-line clinical trial had an increased 9-year CI of non-fatal cardiac events.

Among 240 patients treated with non-intensive approaches, the CI of fatal events was 2.2% at 6 months and 6.8% at 9 years; the CI of non-fatal events was 45.2% at 6 months and 59.6% at 9 years (Appendix A).

### 3.6. QTc Prolongation Events in the Overall Cohort

The 9-year CI of QTcF prolongations was 11.2% for grades 1–2 and 2.7% for grade 3, and no patient had grades 4 or 5. Among 73 patients who developed QT prolongation, 25 (34%) patients developed them during induction, and 20 (27%) developed them during further non-intensive cycles (Appendix A). In the univariate analysis, patients aged ≥65 years, with relevant cardiac antecedents included in the front-line clinical trial or receiving upfront FLT3 inhibitors had an increased 9-year CI of grade 1–2 QTcF prolongations (Appendix A). No significant associations were found for grade 3 QTcF prolongations.

### 3.7. Arrhythmia Events in the Overall Cohort

The 9-year CI of grade 1–2 arrhythmia was 2.3%, grade 3–4 was 9.2%, and grade 5 was 1% (six patients died due to suspected or confirmed arrhythmia). Among 64 patients who developed arrhythmias, 29 (45%) did during induction, and 14 (22%) did during non-intensive further cycles (Appendix A). In the univariate analysis, patients aged ≥ 65 years with ECOG > 1 had an increased 9-year CI of grade 3–4 arrhythmias (Appendix A).

### 3.8. Heart Failure and Myocardial Ischemic Events in the Overall Cohort

The 9-year CI of heart failure events was 1.4% for grades 1–2, 15.3% for grades 3–4, and 2.4% for grade 5 (11 patients died due to heart failure). Among 97 patients who developed cardiac failure events, 52 (54%) patients did during induction, 10 (11%) did during the pre-treatment period, and 13 (15%) did during the post-treatment follow-up period (Appendix A). In the univariate analysis, patients with ECOG > 1 had an increased 9-year CI of grade 3–4 heart failure, and patients with relevant cardiac antecedents had an increased 9-year CI of grade 5 heart failure (Appendix A). Among these 97 patients, 56 (57.7%) had a concomitant infection (i.e., sepsis or pneumonia), and in 24 (24.7%), AML was not in remission.

The 9-year CI of myocardial ischemic events was 2.1% for grades 1–2 and 4.1% for grades 3–4, and no patient had grade 5 (Appendix A).

### 3.9. Clinical Outcomes after Intensive Approaches According to the Development of Cardiac Events

The median OS in the overall cohort was 11 months (10–13 months, CI 95%). The CR + CRi rate after intensive induction was not different between those developing or not developing cardiac events (Table 5). Among 285 patients treated with intensive approaches, the median OS was 22 months (16–38 months, CI 95%): 34 months (18 months–NA (not available), CI 95%) in patients without cardiac events (n = 125), 43 months (22 months–NA, CI 95%) with grade 1–2 (n = 52), 15 months (13–22 months, CI 95%) with grade 3–4 (n = 98), and 5 months (3 months–NA, CI 95%) with grade 5 (n = 10) (*p* < 0.001) (Figure 3A). The median EFS was 10 months (6–13 months, CI 95%): 12 months (10–13 months, CI 95%) in patients without cardiac events, 14 months (6–NA months, CI 95%) in patients with grade 1–2, 8 months (4–13 months, CI 95%) in patients with grade 3–4, and 4 months (3-NA months, CI 95%) in patients with grade 5 (*p* = 0.015) (Figure 3B).

### 3.10. Cardiac Events in the 1L Cohort

When censoring patients at first R/R episode (1L cohort), seven patients (1.3%) experienced fatal cardiac events during the observation period, with a CI of 1.2% at 6 months and 2.8% at 9 years (Figure 1C). In the 1L cohort, 222 (42.3%) patients experienced non-fatal cardiac events, with a CI of 37.8% at 6 months and 43.3% at 9 years (Figure 1D). In the univariate analysis, patients with relevant cardiac antecedents (CI at 9 years, 7.4% vs. 1.9%; *p* < 0.001) and prior exposure to anthracyclines (CI at 9 years, 7.9% vs. 2.4%; *p* = 0.005) had an increased CI of fatal cardiac events. Patients aged ≥ 65 years (49.4% vs. 37.2%, *p* = 0.001), with relevant cardiac antecedents (61% vs. 40.5%, *p* = 0.002), and included in front-line clinical trials (51.8% vs. 36.6%, *p* < 0.001) had an increased CI of non-fatal cardiac events at 9 years (Appendix A). Multivariate analysis showed that patients aged ≥ 65 years (HR 1.8, *p* = 0.007), with relevant cardiac antecedents (HR 1.9, *p* < 0.001), and undergoing non-intensive chemotherapy (HR 1.9, *p* = 0.005) were associated with an increased CI of non-fatal cardiac events (Appendix A).

The CI of grade 1–2 QTcF prolongations was 7.8% at 6 months and 9.2% at 9 years, grade 3 was 1.3% at 6 months and 2% at 9 years, and no patient had grades 4 or 5. In the univariate analysis, patients aged > 65 years (12% vs. 6.4%, *p* = 0.038), those with relevant cardiac antecedents (16.5% vs. 8.1%, *p* = 0.018), included in front-line clinical trials (15.3 vs. 4.8, *p* < 0.001), and treated with FLT3 inhibitors (18.4% vs. 8.4%, *p* = 0.027) had an increased 9-year CI of grade 1–2 QTcF prolongations.

The 9-year CI of grade 1–2 heart failure events was 1%, for grades 3–4 it was 11.1%, and it was 0.4% for grade 5 (two patients died due to heart failure). In the univariate analysis, patients with relevant cardiac antecedents (19.4% vs. 9.7%, *p* = 0.017) and with ECOG > 1 (20.6% vs. 8.6%, *p* < 0.001) had an increased 9-year CI of grade 3–4 heart failure.

The 9-year CI of grade 1–2 arrhythmia was 2%, grade 3–4 was 5.7%, and grade 5 was 0.6% (four patients died due to arrhythmia). In the univariate analysis, patients with relevant cardiac antecedents (12.3% vs. 4.7%, *p* = 0.009) and an ECOG > 1 (12.2% vs. 4.1%, *p* = 0.001) had an increased 9-year CI of grade 3–4 arrhythmias.

## 4. Discussion

Our study shows a high incidence of cardiac events (58.5%) in a real-world series of patients undergoing AML therapy, coupled with significant mortality due to cardiotoxicity. We also show decreased OS and EFS in fit patients developing grade 3–4 cardiotoxic events, while grade 1–2 cardiac events apparently did not impact survival. The presence of relevant cardiac antecedents was the main risk factor for developing fatal cardiac events.

So far, no previous studies have provided a holistic view of cardiac issues in a real-world series of newly diagnosed AML patients followed-up until death or the last patient visit. As a matter of fact, prior studies analyzing cardiac toxicity in AML patients were mainly focused on anthracycline-derived decreases in left ventricular ejection fractions (LVEFs) in adult [7,22] or pediatric [23] patients and/or APL patients [24]. In our study, we analyzed a large series of adult patients with AML, and we focused on all types of cardiac toxicities, not restricted to potential anthracycline-related LVEF decreases. We should highlight that, although this is a real-life study, all patients were treated in a reference institution where a sizable proportion of patients were enrolled in clinical trials in the front line (44%).

Our main objective was to assess the incidence of fatal cardiac events in the overall cohort, and we found an incidence of 4.6% for cardiac-related deaths, rising to 15% among the small percentage (8%) of patients receiving only supportive care. This reflects that some patients cannot start antileukemic therapy due to early death (for instance, due to cardiac events) but also due to comorbidities (such as cardiac comorbidities). When focusing on 525 patients receiving front-line therapy, we observed an estimated CI of 2% at 6 months, which increased to 6.7% at 9 years, reflecting that late fatal cardiac events are frequently occurring among AML patients (due to late toxicities or subsequent treatment lines for R/R episodes). Three out of six patients categorized as having fatal arrhythmias died late, unrelated to front-line therapy (two of them with active diseases), and were recorded as having cardiac arrests.

The estimated cumulative incidence of all types of non-fatal cardiac events was unexpectedly high (57% at 9 years), with a 44% incidence at 6 months, meaning that non-fatal cardiac events occurred early during the treatment phase, with a plateau after 2 years since AML diagnosis. Thus, once patients were successfully treated and off therapy, non-fatal cardiac events were rare, probably due to less exposure to toxic agents, but also reflecting less medical monitoring and reporting. As expected, most of the cardiac events occurred during induction therapy, but post-remission cycles and post-transplant periods were also accompanied by a significant number of events. Most patients in our study developed their first events during the induction or the first cycle, probably due to uncontrolled AML, fluid overload, and severe infections such as pneumonia and sepsis. In fact, we observed that more than half of patients developing a heart failure event had a concomitant relevant infection, in line with the results of a COG AAML0531 pediatric study [23]. Another relevant finding is that, according to CTCAE, most cardiac events were grade ≥ 3, and this was inherent to the CTCAE definitions [18]. To assess the true severity and seriousness of cardiac events, we analyzed the incidence of so-called life-threatening events, resulting in a substantial fatal plus life-threatening crude incidence of 8.7%.

Our study included a detailed analysis of the occurrence of QT prolongation, and as far as we know, this is the largest real-life AML study focusing on this issue. The frequency of all grades of QTcF prolongation in a front-line phase 3 clinical trial with FLT3-inhibitor quizartinib (QuANTUM-First) was 34% in the quizartinib arm and 18% in the control arm. In our analysis, 21% of patients included in front-line clinical trials (the vast majority without FLT3 inhibitors) developed any grade of QTcF prolongation, such as the control arm of the QuANTUM-First trial [25]. In our series, grade 3 QTcF was infrequent (around 2%) in patients either receiving or not receiving FLT3 inhibitors whether included or not in clinical trials, similar to the Quantum-First trial (2% in the quizartinib arm and 1% in the control arm). We can speculate that the higher incidence of QT prolongation among those of our patients who were included in trials was driven not only by the experimental agents’ toxicity but also, at least in part, by the tight ECG monitoring, as per protocol safety procedures. Importantly, in our series, no patient developed a death related to QT prolongation, as per physician judgment.

Heart failure grade 1–2 events were rare, while grade 3–4 events occurred in 12.2% at 6 months and 15.3% at 9 years. It should be noted that most heart failures occurred during induction—coinciding with exposure to anthracyclines and other cardiotoxic agents—with active AML and with the potential co-occurrence of infectious episodes that can trigger some cases into cardiac failure. Interestingly, a COG AAML0531 pediatric study including almost 900 AML patients analyzed the incidence of heart failure (defined as an LVEF decrease) using echocardiographic monitoring. In line with our results, they identified LVEF decreases in 12% of patients during follow-up, 70% of whom experienced onset during the on-protocol period, and a sizable proportion developed heart failure coinciding with sepsis [23]. As serial echocardiogram monitoring was not performed in our study, some low-LVEF events may have been missed, especially in asymptomatic patients. In this regard, a study by Rodríguez-Veiga et al. analyzed late cardiomyopathy via longitudinal assessment using magnetic resonance imaging in 82 patients with acute promyelocytic leukemia treated with anthracycline-containing regimens, showing 12% had developed subclinical cardiomyopathy [26].

In our study, the majority of cardiac events occurred during induction and active therapy phases, suggesting that the late cardiotoxic effects of anthracyclines might be relatively rare. Nevertheless, prior anthracycline exposure is a known factor for developing cardiac toxicity, and we found that this was a risk factor for fatal cardiac events in the univariate analysis when restricted to the first-line cohort. However, we observed increased cardiotoxicity among unfit patients treated without anthracyclines, especially in those with prior cardiac antecedents.

We show that the development of grade 3–4 cardiac events (among patients treated intensively) impacted the OS and EFS, as reported by the COG study [23]. However, we show that the development of isolated grade 1–2 cardiac toxicity had no impact on survival outcomes. Our data highlight that there might be some concern when patients develop cardiac toxicities limited to mild severity, as those seem to be irrelevant to patient outcomes.

There are several studies focused on cardio-protection after treatment with anthracyclines via the use of dexrazoxane, which has been recommended in children since the beginning of anthracycline treatment and in adults after reaching anthracycline doses of 300 mg/m^2^ [27,28]. Other strategies include the use of liposomal products, angiotensin-converting enzyme (ACE) inhibitors, spironolactone, and β-blockers. A recent systematic review focused mostly on adult patients receiving cardiotoxic drugs (the majority included anthracyclines) found that spironolactone followed by enalapril, nebivolol, and statins was associated with the greatest LVEF improvement, and patients treated with enalapril had a lower risk of developing clinical heart failure [29,30]. These strategies could be applied to patients at a higher risk of developing cardiac events, as found in our study.

We also analyzed cardiac events restricted to the first line (censoring at relapse or refractory status). Interestingly, the rate of all cardiac grade events in 1L patients undergoing intensive chemotherapy was 42.7%, similar to the 46% reported by investigators of a phase 2 206 study assessing the effects of the cardiac repolarization of CPX-351 on patients with acute leukemias [31]. It should be noted that cardiac events were seen in 20% of patients receiving venetoclax with hypomethylating agents for the 1L treatment of AML in 170 less-fit patients treated at the Mayo Clinic [32]. Most events occurred during treatment cycles 1 (41%) and 2 (26%), and 9 patients died due to a cardiac event (5.2%). We can speculate on several reasons explaining the higher incidence of cardiac events in our study: (1) cardiac events were assessed only while treatment was ongoing, so less follow-up was applied; (2) patients were treated outside the context of a clinical trial, and less tight monitoring, including systematic ECG, was probably performed.

Recently, cardiovascular disease has been associated with clonal hematopoiesis, especially *TET2* mutations [33,34]. In our institution, routine NGS at diagnosis has been performed since 2017, so we were not able to analyze the association between cardiac events and different mutations in leukemic cells other than *FLT3-ITD*. On the other hand, it has been hypothesized that genetic variability in genes involved in the anthracycline metabolic pathway could be one of the causes of the differences in clinical outcomes and toxicities such as cardiotoxicity [35]. In a previous study performed on 225 adult de novo AML patients at our institution, we found that a combination of *SLC22A16* rs714368 and *ABCG2* rs2231142 polymorphisms was related to cardiac toxicity [36].

Although our study is limited by a single-institution retrospective design and baseline echocardiograms or ECGs were not obtained for all patients, we were focused on showing a real-world evidence study. Importantly, our data were not collected in the context of classical clinical trial monitoring, but data were captured and reviewed by experienced hematologists able to medically and homogeneously interpret data from medical records.

## 5. Conclusions

In summary, our study shows a high incidence of cardiac events through patient journeys in a large, real-world series of 525 patients undergoing AML therapy, with significant mortality due to cardiotoxicity. We have identified prognostic factors that increase the risk of cardiac events, which may be useful in selecting high-risk patients for stringent cardiac monitoring and management. The inclusion of patients in clinical trials could lead to an increased observed incidence of non-fatal cardiac events, probably due to the use of novel cardiotoxic agents, but also due to the inherent tight monitoring of these patients leading to the documentation of cardiac events.

## Figures and Tables

**Figure 1 cancers-15-02267-f001:**
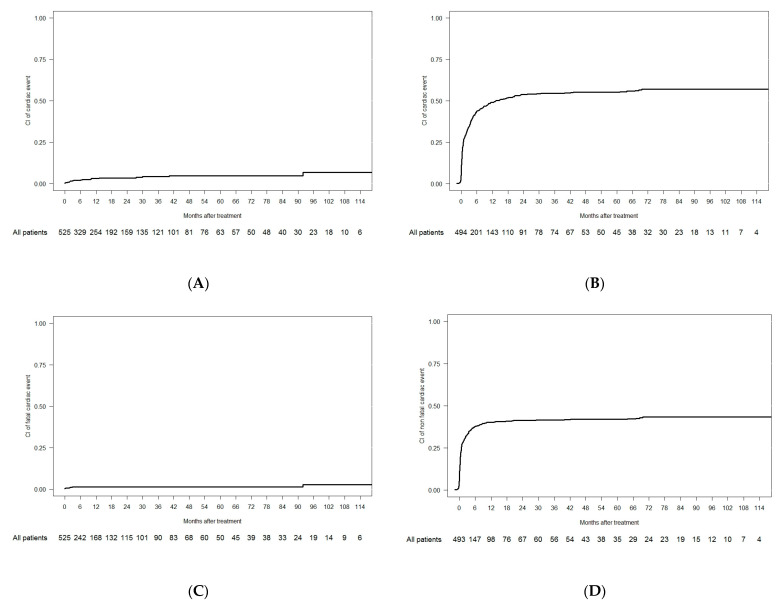
Cumulative incidence of cardiac events. (**A**) CI of fatal cardiac events in the overall cohort; (**B**) CI of non-fatal cardiac events in the overall cohort; (**C**) CI of fatal cardiac events in the 1L cohort; (**D**) CI of non-fatal cardiac events in the 1L cohort.

**Figure 2 cancers-15-02267-f002:**
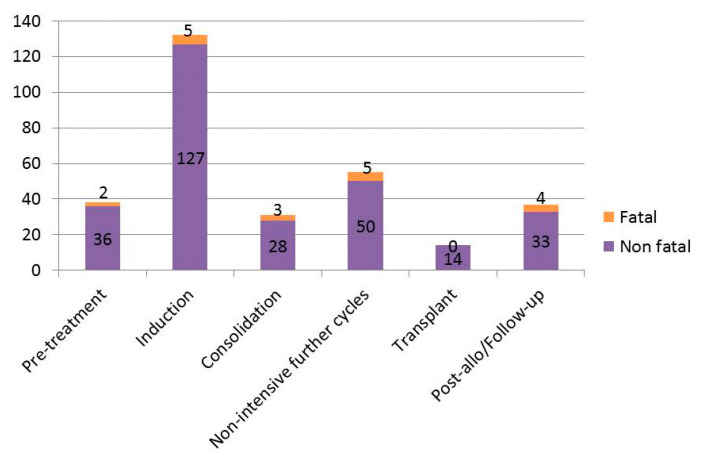
Timing of cardiac events in the overall cohort.

**Figure 3 cancers-15-02267-f003:**
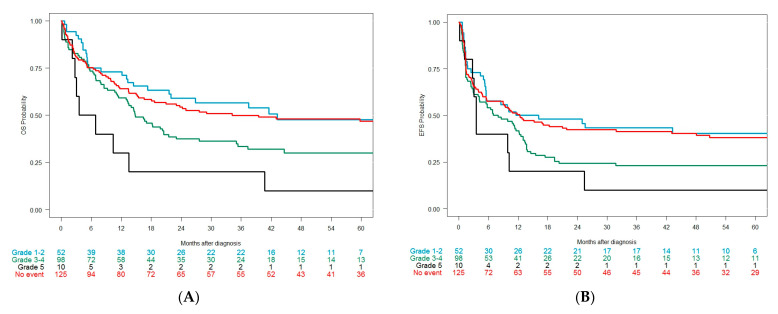
(**A**) Overall survival in the overall cohort (intensive chemotherapy only) according to the grade of the cardiac event; (**B**) event-free survival in the overall cohort according to the grade of the cardiac event.

**Table 1 cancers-15-02267-t001:** Front-line therapeutic approach in AML patients (intensive vs. non-intensive vs. trial vs. best supportive care).

Therapeutic Approach	Schedule	Number of Patients n = 571 (%)
**Intensive chemotherapy**		218 (38)
	IDA ^1^ + Ara-C ^2^ (3 + 7)	195 (34)
	IDA + Ara-C (2 + 5)	3 (1)
	FLAG-IDA (fludarabine + Ara-C + IDA)	10 (2)
	IDA + Ara-C (3 + 7) + midostaurin	4 (1)
	Allogeneic transplant	4 (1)
	Other intensive chemotherapy	2 (0.3)
**Non-intensive therapy**		75 (13)
	Azacitidine	6 (1)
	Decitabine	1 (0.2)
	FLUGA (fludarabine + LD-Ara-C)	68 (12)
**Clinical Trial**		232 (41)
**Intensive**		67 (12)
	Intensive + FLT3 inhibitor/ placebo	31 (5)
	Intensive without FLT3 inhibitor	36 (6)
**Non-intensive**		165 (29)
	Non-intensive + FLT3 inhibitor	1 (0.2)
	Non-intensive without FLT3 inhibitor	164 (29)
**Supportive care only**		46 (8)

^1^ IDA: idarubicina; ^2^ Ara-C: cytarabine.

**Table 2 cancers-15-02267-t002:** Demographic, clinical, and biological characteristics of patients at diagnosis according to previous cardiac comorbidities.

Characteristic	Overall		Relevant Cardiac Comorbidities		No Cardiac Comorbidities		*p*
	Median (Range)	n (%)	Median (Range)	n (%)	Median (Range)	n (%)	
**N**		571 (100)		82 (14)		489 (86)	
**Age, years**	65 (18–98)		72 (37–98)		64 (18–92)		<0.001 *
<65		272 (48)		25 (30)		247 (51)	0.001
≥65		299 (52)		57 (70)		242 (49)	
**Gender**		571		82		489	
Male		331 (58)		60 (73)		271 (55)	0.004
Female		240 (42)		22 (27)		218 (45)	
**ECOG**	1 (0–4)	571	1 (0–4)	82	1 (0–4)	489	0.001 *
0–1		432 (76)		54 (66)		378 (77)	0.04
≥2		139 (24)		28 (34)		111 (23)	
**Comorbidities**		571		82		489	
Yes		379 (66)		75 (91)		304 (62)	<0.001
No		192 (34)		7 (9)		185 (38)	
**Type of AML**		571		82		489	
De novo		350 (61)		45 (55)		305 (62)	0.298
Therapy-related		91 (16)		13 (16)		78 (16)	
Previous MDS/MPN		130 (23)		24 (29)		106 (22)	
**WHO classification**		571		82		489	
AML-RGA		154 (27)		20 (24)		134 (27)	0.819
AML-NOS		68 (12)		10 (12)		58 (12)	
AML-MRC		240 (42)		41 (50)		199 (41)	
t-AML		52 (9)		5 (6)		47 (10)	
Myeloid sarcoma		2 (0.4)		0		2 (0.4)	
Ambiguous lineage		8 (1)		1 (1)		7 (1)	
BPDCN		2 (0.4)		0		2 (0.4)	
Not available		45 (8)		5 (6)		40 (8)	
**Extramedullary disease**		569		82		487	
Yes		100 (18)		14 (17)		86 (18)	0.978
No		469 (82)		68 (83)		401 (82)	
**WBC, ×10^9^/L**	8 (0.3–434.3)	571	8.2 (0.3–300.2)	82	8 (0.3–434.3)	489	0.640 *
≤5		241 (42)		35 (43)		206 (42)	0.705
5–10		68 (12)		11 (13)		57 (12)	
10–50		153 (27)		24 (29)		129 (26)	
> 50		109 (19)		12 (15)		97 (20)	
**Hemoglobin, g/dL**	8.8 (2.9–15.5)	571	8.4 (5–12.9)	82	8.8 (2.9–15.5)	489	0.078 *
≤10		434 (76)		70 (85)		364 (74)	0.044
>10		137 (24)		12 (15)		125 (26)	
**Platelet count, ×10^9^/L**	53 (1–1442)	570	47 (8–816)	81	54 (1–1442)	489	0.275 *
≤20		109 (19)		16 (20)		93 (19)	0.997
> 20		461 (81)		65 (80)		396 (81)	
**PB blasts, %**		563		82		481	
≤50		417 (74)		64 (78)		353 (73)	0.451
> 50		146 (26)		18 (22)		128 (27)	
**BM blasts, %**	47 (0–100)	559	43 (14–100)	81	48 (0–100)	478	0.239 *
≤30		158 (28)		28 (35)		130 (27)	0.339
>30≤70		234 (42)		29 (36)		205 (43)	
>70		167 (30)		24 (30)		143 (30)	
**Creatinine, mg/dL**	0.9 (0.1–7.4)	570	1 (0.2–5.4)	82	0.8 (0.1–7.4)	488	<0.001 *
≤1.3		482 (85)		58 (71)		424 (87)	<0.001
> 1.3		88 (15)		24 (29)		64 (13)	
**Urea, mg/dL**	36 (6–174)	554	45 (18–158)	81	35 (6–174)	473	<0.001 *
≤50		429 (77)		46 (57)		383 (81)	<0.001
>50		125 (23)		35 (43)		90 (19)	
**Uric acid, mg/dL**	5 (1–29)	489	5.8 (1.4–17.1)	68	4.9 (1–29)	421	0.002 *
≤7		398 (81)		48 (71)		350 (83)	0.022
>7		91 (19)		20 (29)		71 (17)	
**Bilirubin, mg/dL**	0.6 (0.1–6.2)	553	0.7 (0.1–2.3)	77	0.6 (0.1–6.2)	472	0.313 *
≤1.2		484 (88)		68 (86)		416 (88)	0.813
>1.2		69 (12)		11 (14)		58 (12)	
**AST, U/L**	22 (6–1085)	549	22 (7–185)	77	22 (6–1085)	472	0.061 *
≤50		482 (88)		69 (90)		413 (88)	0.821
>50		66 (12)		8 (10)		58 (12)	
**ALT, U/L**	18 (3–714)	566	16 (3–125)	80	19 (3–714)	486	0.68 *
≤50		504 (89)		75 (94)		429 (88)	0.207
>50		62 (11)		5 (6)		57 (12)	
**Albumin, g/dL**	3.7 (1.6–5.2)	518	3.7 (2.2–4.8)	69	3.7 (1.6–5.2)	449	0.148 *
≤3.5		204 (39)		31 (45)		173 (39)	0.379
>3.5		314 (61)		38 (55)		276 (61)	
**LDH, U/L**	521 (101–42630)	558	627 (149–11800)		509 (101–42630)		0.984 *
≤600		307 (55)		38 (48)		269 (56)	0.181
>600		251 (45)		42 (53)		209 (44)	
**Fibrinogen**	487 (34–1150)	548	493 (162–1002)	79	487 (34–1150)	469	0.947 *
≤170		6 (1)		1 (1)		5 (1)	0.67
>170		542 (99)		78 (99)		464 (99)	
**Prothrombin time**		501		70		431	
Prolonged		175 (35)		33 (47)		142 (33)	0.03
Normal		326 (65)		37 (53)		289 (67)	
**APTT**		541		76		465	
Prolonged		73 (13)		10 (13)		63 (14)	0.917
Normal		468 (87)		66 (87)		401 (86)	
**Cytogenetics**		571		82		489	
Normal		211 (37)		34 (41)		177 (36)	0.8
Abnormal		293 (51)		38 (46)		255 (52)	
No metaphases		47 (8)		7 (9)		40 (8)	
Not available		20 (4)		3 (4)		17 (3)	
**MRC Cytogenetic risk**		517		74		443	
Favorable		27 (5)		1 (1)		26 (6)	0.185
Intermediate		307 (59)		49 (66)		258 (58)	
Adverse		183 (35)		24 (32)		159 (36)	
** *FLT3-ITD* **		517		77		440	
Positive		79 (15)		9 (12)		70 (16)	0.437
Negative		438 (85)		68 (88)		370 (84)	
***FLT3-ITD*** **ratio**		517		77		440	
<0.05		452 (87)		70 (91)		382 (87)	0.752
0.05–0.5		24 (5)		2 (3)		22 (5)	
0.5–0.8		23 (4)		3 (4)		20 (5)	
≥0.8		18 (3)		2 (3)		16 (4)	
** *FLT3-TKD* **		505		75		430	
Positive		26 (13)		6 (8)		20 (5)	0.217
Negative		479 (93)		69 (92)		410 (93)	
** *NPM1* **		514		77		437	
Positive		119 (23)		18 (23)		101 (23)	0.924
Negative		395 (77)		59 (77)		336 (77)	
** *CEBPA* **		571		82		489	
Positive		14 (2)		2 (2)		12 (2)	0.427
Negative		317 (56)		47 (57)		270 (55)	
Not available		240 (42)		33 (40)		207 (42)	
** *IDH* **		571		82		489	
*IDH1* positive		23 (4)		3 (4)		20 (4)	0.67
*IDH2* positive		54 (9)		6 (7)		48 (10)	
Negative		266 (47)		43 (52)		223 (46)	
Not available		228 (40)		30 (37)		198 (40)	
**Therapeutic approach**		571		82		489	
Intensive		218 (38)		19 (23)		199 (41)	0.043
HMA		7 (1)		2 (2)		5 (1)	
LDAC-based		68 (12)		11 (13)		57 (12)	
Clinical trial, intensive		67 (12)		9 (11)		58 (12)	
Clinical trial, non-intensive		165 (29)		32 (39)		133 (27)	
BSC		46 (8)		9 (11)		37 (8)	

* *p* compares continuous variables.

**Table 3 cancers-15-02267-t003:** Overall cohort: Crude and cumulative incidence of cardiac events (fatal vs. non-fatal vs. no cardiac event) according to demographic, clinical, and biological characteristics of patients at diagnosis.

	No Cardiac event	*p*-Value *	Fatal Cardiac Event				Non-Fatal Cardiac Event			
Characteristic	N (%)		Crude incidence N (%)	Cumulative Incidence			Crude incidence N (%)	Cumulative Incidence		
				At 6 months, %	At Last FU, %	*p*		At 6 months, %	At Last FU, %	*p*
**N**	218 (42)		19 * (3.6)	2	6.7		288 (54.9)	43.7	56.9	
**Age, N**	218		19				288			
**<65 years**	128 (48.5)	0.005	9 (3.4)	1.6	5.9	0.25	127 (48.1)	37.8	49.9	<0.001
**≥65 years**	90 (34.5)		10 (3.8)	2.5	6.8		161 (61.7)	49.7	64.4	
**Relevant cardiologic antecedents**	218		19				288			
**No**	202 (44.7)	<0.001	10 (2.2)	1.2	4.9	<0.001	240 (53.1)	41.8	54.6	0.004
**Yes**	16 (21.9)		9 (12.3)	7.2	20.1		48 (65.8)	56.4	73	
**All cardiologic antecedents**	218		19				288			
**No**	163 (49.7)	<0.001	6 (1.8)	0.7	4.1	0.0004	159 (48.5)	37.6	49.7	<0.001
**Yes**	55 (27.9)		13 (6.6)	4.4	10.7		129 (65.5)	54.2	70.6	
**Previous anthracycline treatment**	218		19				288			
**No**	207 (42.1)	0.51	17 (3.5)	1.7	6.5	0.22	268 (54.5)	43.3	56.3	0.49
**Yes**	11 (33.3)		2 (6.1)	6.7	6.7		20 (60.6)	50.5	60.7	
**ECOG at diagnosis**	218		19				288			
**<2**	173 (41.4)	0.99	15 (3.6)	2	6.7	0.56	230 (55)	43.3	57.2	0.63
**≥2**	45 (42.1)		4 (3.7)	1.9	5.5		58 (54.2)	45.5	56.1	
***FLT3-ITD*** **status**	197		19				269			
**Negative**	161 (39.5)	0.44	17 (4.2)	2.3	7.8	0.53	230 (56.4)	44.2	58.8	0.52
**Positive**	36 (46.8)		2 (2.6)	1.4	3.1		39 (50.7)	43.1	51.3	
**Treatment approach**	218		19				288			
**Intensive**	125 (43.9)	0.5	10 (3.5)	1.9	6.3	0.37	150 (52.6)	42.5	54.7	0.17
**Non-intensive**	93 (38.8)		9 (3.8)	2.2	6.8		138 (57.5)	45.2	59.6	
**Inclusion in clinical trial**	218		19				288			
**No**	140 (47.8)	0.004	8 (2.7)	1.1	5.4	0.06	145 (49.5)	37.4	50.8	<0.001
**Yes**	78 (33.6)		11 (4.7)	3.2	6.5		143 (61.6)	51.8	65.2	
**Use of FLT3 inhibitors**	218		19				288			
**No**	204 (41.9)	0.75	18 (3.7)	2.2	4.8	0.76	265 (54.4)	42.8	56.3	0.28
**Yes**	14 (36.8)		1 (2.6)	0	3.1		23 (60.5)	55.3	60.7	

* This *p*-value compares the crude incidence between the three groups. FU: follow-up

**Table 4 cancers-15-02267-t004:** Overall cohort: Crude and cumulative incidence of cardiac events (fatal vs. life-threatening vs. no cardiac event) according to demographic, clinical, and biological characteristics of patients at diagnosis.

	No Cardiac Event	*p*-Value *	Non-Life-Threatening				Life-Threatening				Fatal			
Characteristic	N (%)		Crude Incidence N (%)	Cumulative Incidence			Crude Incidence N (%)	Cumulative Incidence			Crude Incidence N (%)	Cumulative Incidence		
				At 6 months, %	At Last FU, %	*p*		At 6 months, %	At Last FU, %	*p*		At 6 months, %	At Last FU, %	*p*
**N**	218 (41.5)		261 (49.7)	40	53.1		27 (5.1)	4.1	6.4		19 (3.6)	2	6.7	
**Age**	218		261				27				19			
**<65 years**	128 (48.5)	<0.001	108 (40.9)	32.9	43.8	<0.001	19 (7.2)	4.7	8.2	0.11	9 (3.4)	1.6	5.9	0.25
**≥65 years**	90 (34.5)		153 (58.6)	47.2	62.9		8 (3.1)	3.5	3.5		10 (3.8)	2.5	6.8	
**Relevant cardiologic antecedents**	218		261				27				19			
**No**	202 (44.7)	<0.001	215 (47.6)	37.9	50.4	0.001	25 (5.5)	4.4	6.6	0.45	10 (2.2)	1.2	4.9	<0.001
**Yes**	16 (21.9)		46 (63)	53.4	71.7		2 (2.7)	1.9	4		9 (12.3)	7.2	20.1	
**Previous anthracycline treatment**	218		261				27				19			
**No**	207 (42.1)	0.6	242 (49.2)	39.5	52.3	0.33	26 (5.3)	4.1	6.5	0.7	17 (3.5)	1.7	6.5	0.22
**Yes**	11 (33.3)		19 (57.6)	47.5	58.8		1 (3)	4	4		2 (6.1)	6.7	6.7	
**ECOG at diagnosis**	218		261				27				19			
**<2**	173 (41.4)	0.99	209 (50)	39.6	53.3	0.6	21 (5)	3.8	6.1	0.56	15 (3.6)	2	6.7	0.56
**≥2**	45 (42.1)		52 (48.6)	41.3	52.6		6 (5.6)	5.5	7.5		4 (3.7)	1.9	5.5	
**FLT3-ITD status**	197		243				26				19			
**Negative**	161 (39.5)	0.55	209 (51.2)	40.3	55.3	0.34	21 (5.2)	4.2	6.1	0.63	17 (4.2)	2.3	7.8	0.53
**Positive**	36 (46.8)		34 (44.2)	38.2	45.5		5 (6.5)	5.6	7.3		2 (2.6)	1.4	3.1	
**Treatment chemotherapy**	218		261				27				19			
**Intensive**	125 (43.9)	0.07	130 (45.6)	37.6	48.8	0.025	20 (7)	4.8	8.3	0.088	10 (3.5)	1.9	6.3	0.37
**Non-intensive**	93 (38.8)		131 (54.6)	42.7	58.1		7 (2.9)	3.2	3.2		9 (3.8)	2.2	6.8	
**Inclusion in clinical trial**	218		261				27				19			
**No**	140 (47.8)	<0.001	125 (42.7)	32.8	45.4	<0.001	20 (6.8)	5	8	0.096	8 (2.7)	1.1	5.4	0.06
**Yes**	78 (33.6)		136 (58.6)	49.2	63.5		7 (3)	2.9	3.5		11 (4.7)	3.2	6.5	
**Use of FLT3 inhibitors**	218	0.71	261				27				19			
**No**	204 (41.9)		239 (49.1)	38.9	52.3	0.19	26 (5.3)	4.2	6.7	0.43	18 (3.7)	2.2	4.8	0.76
**Yes**	14 (36.8)		22 (57.9)	53	58.8		1 (2.6)	2.7	2.7		1 (2.6)	0	3.1	

* This *p*-value compares the crude incidence between the four groups. FU: follow-up.

**Table 5 cancers-15-02267-t005:** Main clinical outcomes (CR/CRi/OS/EFS) in intensive chemotherapy AML patients according to the occurrence of their first cardiac events (grade 0 vs. 1–2 vs. 3–4 vs. 5) from index date to last follow-up.

	All Patients	Cardiac Event			No Cardiac Event	*p*
		Grades 1–2	Grades 3–4	Grade 5		
	N (%)	N (%)	N (%)	N (%)	N (%)	
**Response, n (%)**	285 (100)	52 (100)	98 (100)	10 (100)	125 (100)	
**ORR (CR + CRi)**	193 (67.7)	39 (75)	62 (63.3)	8 (80)	84 (67.2)	0.426
**CR**	184 (64.6)	35 (67.3)	58 (59.2)	7 (70)	84 (67.2)	
**CRi**	9 (3.1)	4 (7.7)	4 (4.1)	1 (10)	0 (0)	
**PR**	12 (4.2)	1 (1.9)	4 (4.1)	0 (0)	7 (5.6)	
**Resistance**	50 (17.5)	10 (19.2)	19 (19.4)	1 (10)	20 (16)	
**Induction death**	30 (10.5)	2 (3.8)	13 (13.3)	1 (10)	14 (11.2)	
**OS, n (%)**						
**Median (CI95), months**	21.6 (15.5–37.5)	43.2 (21.6-NA)	14.8 (13–21.5)	5.2 (2.8-NA)	34.2 (18-NA)	<0.001
**1 year (CI95), %**	63 (57–68)	73 (62–86)	59 (50–70)	30 (12–77)	64 (56–73)	
**2 years (CI95), %**	48 (42–54)	59 (47–74)	38 (29–49)	20 (6–69)	64 (46–64)	
**3 years (CI95), %**	44 (39–50)	57 (44–72)	34 (25–45)	20 (6–69)	50 (42–60)	
**5 years (CI95), %**	39 (34–46)	48 (35–65)	30 (22–42)	10 (2–64)	47 (39–57)	
**EFS, n (%)**						
**Median (CI95), months**	10.2 (6.4–13)	13.9 (5.5-NA)	7.5 (4.1–12.6)	3.5 (2.8-NA)	12.3 (5.7–48)	0.015
**1 year (CI95), %**	46 (41–52)	50 (38–66)	42 (33–53)	20 (6–69)	50 (42–60)	
**2 years (CI95), %**	36 (31–42)	48 (36–64)	24 (17–35)	20 (6–69)	42 (35–52)	
**3 years (CI95), %**	34 (29–40)	43 (32–60)	23 (16–33)	10 (2–64)	41 (34–51)	
**5 years (CI95), %**	32 (26–38)	40 (29–57)	23 (16–33)	10 (2–64)	38 (30–48)	

ORR: overall response rate; OS: overall survival; EFS: event-free survival.

## Data Availability

The data presented in this study are available in this article and Appendix A.

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
