# Peer review of "Incidence and Risk Factors for Development of Cardiac Toxicity in Adult Patients with Newly Diagnosed Acute Myeloid Leukemia"

_cancers, 2023, doi:10.3390/cancers15082267_

Round 1

Reviewer 1 Report

Cancers-2253885

Incidence and Risk Factors for Development of Cardiac Toxicity in Adult Patients with Newly Diagnosed Acute Myeloid Leukemia 

The article “Incidence and Risk Factors for Development of Cardiac Toxicity in Adult Patients with Newly Diagnosed Acute Myeloid Leukemia (cancers-2253885)” by Boluda B, et al. demonstrated that the cardiac toxicity was not rare, and associated with poor clinical outcome in the patients with AML. The risk factors, including their onset, were analyzed very well. This retrospective study was very interesting, and have several important suggestions for clinical practice. Therefore, I considered that this article was suitable for acceptance of Cancers. However, I have several comments to improve this article.

1.       The majority of cardiac toxicity occurred during induction therapy, suggesting that the anthracycline related cardiac toxicity, especially low ejection function due to high cumulative dose of anthracycline, might be rare in this study. Therefore, the author could analyze the relationship between incidence of cardiac events and anthracycline and discuss about that in discussion part.

2.       Why did the cardiac events occur frequently during the induction therapy? For, instance, generally speaking, severe infection, such as pneumonia and sepsis, and acute kidney injury, including tumor-lysis syndrome, occurred frequently during induction therapy.

3.       In this study, reduced ejection function was defined as one of cardiac events. Do the authors monitor the ejection function regularly? If not, I considered that there was potential bias that the incidence of low EF was more frequently pointed out in the patients with cardiac complications because they might visit the cardiovascular department. Therefore, the authors could add several comment as limitation.

4.       The title of 3.2 was not correct in line. 205. I considered that the correct title was “3.2 fatal cardiac events in the overall cohort” .

Reviewer 2 Report

I have some suggestions in the discussion that could help make the manuscript better. 

1. Add a figure and a few sentences on potential cardioprotective strategies being discussed in the literature. An excellent example of this is provided in Minimizing cardiac toxicity in children with acute myeloid leukemia by Narayan et al (figure 4 for reference). Provide more examples from the literature for potential cardio-protective targets

2. Expand on the timing of cardio-toxicities from results sections 3.4 and 3.5 – Is there any significance for the timing of events as it relates to the phase of treatment? Are there any well-defined models to characterize this?

3. Is there any connection between R/R AML setting and the incidence rate of MACE early vs late in your cohort?

4. From the literature, what genetic factors contribute to a higher incidence rate of cardiotoxicities?

5. The use of captopril has been studied in pediatric AML for reducing the incidence rate of cardiotoxicities. How well has that translated to adults? A few phase III studies have been published on this topic, and please cover them in the discussion section.

Reviewer 3 Report

In line 205 (3.2 title), the first letter is missed. Fatal is written atal.

Author Response

Comments: In line 205 (3.2 title), the first letter is missed. Fatal is written atal.

Response: Done
